# Urinary Biomarkers for Cell Cycle Arrest TIMP-2 and IGFBP7 for Prediction of Graft Function Recovery after Kidney Transplantation

**DOI:** 10.3390/ijms25084169

**Published:** 2024-04-10

**Authors:** Anja Gäckler, Onurcan Ertasoglu, Hana Rohn, Justa Friebus-Kardash, Philipp-Christopher Ickerott, Oliver Witzke, Andreas Kribben, Bruno Vogt, Suzan Dahdal, Spyridon Arampatzis, Ute Eisenberger

**Affiliations:** 1Department of Nephrology, University Medicine Essen, University of Duisburg-Essen, Hufelandstraße 55, 45147 Essen, Germanyjusta.friebus-kardash@uk-essen.de (J.F.-K.); andreas.kribben@uk-essen.de (A.K.); ute.eisenberger@kk-bottrop.de (U.E.); 2Department of Infectious Diseases, West German Centre of Infectious Diseases, University Medicine Essen, University of Duisburg-Essen, Hufelandstraße 55, 45147 Essen, Germany; hana.rohn@uk-essen.de (H.R.); oliver.witzke@uk-essen.de (O.W.); 3Department of Nephrology and Hypertension, Inselspital, Bern University Hospital, University of Bern, Julie-von-Jenner-Haus, Freiburgstraße 15, 3010 Bern, Switzerland; bruno.vogt@insel.ch (B.V.); suzan.dahdal@insel.ch (S.D.); spiros.arampatzis@hin.ch (S.A.)

**Keywords:** TIMP-2, IGFBP7, kidney transplantation, graft function recovery

## Abstract

TIMP-2 and IGFBP7 have been identified and validated for the early detection of renal injury in critically ill patients, but data on recovery of allograft function after kidney transplantation (KTx) are scarce. In a prospective observational multicenter cohort study of renal transplant recipients, urinary [TIMP-2] × [IGFBP7] was evaluated daily from day 1 to 7 after KTx. Different stages of early graft function were defined: immediate graft function (IGF) (decrease ≥ 10% in serum creatinine (s-crea) within 24 h post KTx); slow graft function (SGF) (decrease in s-crea < 10% within 24 h post KTx); and delayed graft function (DGF) (any dialysis needed within the first week after KTx). A total of 186 patients were analyzed. [TIMP-2] × [IGFBP7] was significantly elevated as early as day 1 in patients with DGF compared to SGF and IGF. ROC analysis of [TIMP-2] × [IGFBP7] at day 1 post-transplant for event “Non-DGF” revealed a cut-off value of 0.9 (ng/mL)^2^/1000 with a sensitivity of 87% and a specificity of 71%. The positive predictive value for non-DGF was 93%. [TIMP-2] × [IGFBP7] measured at day 1 after KTx can predict early recovery of transplant function and is therefore a valuable biomarker for clinical decision making.

## 1. Introduction

Delayed or poor graft function puts the outcome after kidney transplantation at risk, affecting short- and long-term graft survival with effects on patients’ morbidity and mortality [1,2]. Reasons for delayed graft function (DGF) are multifactorial, including the use of marginal organs by accepting older or expanded criteria donors [3,4], as well as inflammatory and ischemia-reperfusion-derived injury acquired within the donor, during cold storage and reperfusion, or even ongoing within the recipient, resulting in renal allograft failure [5]. A general definition for DGF does not exist, but the need for dialysis within the first week following transplantation is the main diagnostic parameter used [6,7]. However, not only are patients experiencing the need for dialysis treatment at risk for poor graft function and failure, but also those with slow graft function (SGF) defined by delayed restoration of kidney function without need for dialysis [8,9]. Stratification of renal transplant recipients with regard to renal recovery after ischemia-reperfusion injury can facilitate patient management in the early postoperative phase and potentially help to postpone invasive diagnostic procedures, e.g., a kidney allograft biopsy in case of allograft dysfunction.

Prediction of DGF and SGF using conventional renal function parameters, such as serum creatinine, is difficult due to patients starting within different ranges depending on muscle mass, nutrition, time from last dialysis session to transplantation, and the quality of renal replacement therapy. Moreover, serum creatinine is a marker of renal function rather than of renal damage.

The urinary biomarker combination of tissue inhibitor of metalloproteinase-2 (TIMP-2) and insulin-like growth factor 7 (IGFBP7), both markers for G1 cell cycle arrest, has been identified and validated for the early detection of renal injury in critical ill patients [10]. Data on the applicability of the biomarker combination on the early prediction of renal recovery from ischemia-reperfusion injury are rare. In this prospective, multicenter, observational study, we therefore evaluated whether post-transplant urinary [TIMP-2] × [IGFBP7] can differentiate between DGF, SGF, and immediate graft function (IGF).

## 2. Results

### 2.1. Study Cohort

A total of 186 renal transplant recipients were analyzed out of a cohort of 202 transplant recipients. Sixteen patients had to be excluded due to incomplete data or exposure to methylene blue. We observed DGF in 34 (18.3%), SGF in 43 (23.1%), and IGF in 109 (58.6%) renal transplant recipients (Figure 1). After 1 year, patient survival was 94%, and survival with a functioning graft was 89% without any significant difference between functional groups. Recipient and donor characteristics are summarized in Table 1.

A total of 73 out of 186 renal transplant recipients were female, and 32 (17%) of all patients received a re-transplant. Median recipient age at transplantation was 55 years, with younger recipients in the IGF group. In parallel, the median dialysis vintage time was shorter in the IGF group (20 months) than in the DGF and SGF groups (SGF: 38 months, *p* < 0.05 vs. IGF; DGF: 48 months, *p* < 0.001 vs. IGF). These findings are related to a high rate of living donation in the IGF group (39%), compared to the DGF (9%) and SGF group (5%). Pre-immunization status measured as panel reactive antibodies (PRAs) above 5% was 16% overall, with no relevant differences between all functional groups. Causes of chronic kidney disease are summarized in Table 1. Initial immunosuppressive therapy was mainly based on a triple immunosuppressive regimen including corticosteroids (99%), mycophenolate mofetil or mycophenolic acid (94%), and calcineurin inhibitors (98%). In most patients, IL-2 receptor antagonist-based induction therapy was used; only 4% received an induction therapy with anti-thymocyte globulin (ATG).

The median donor age was 54 years, and 55% of all donors were female. The median cold ischemic time (CIT) was 8.9 h, with a significantly shorter CIT in the IGF group (6.3 h) compared to both of the other groups (DGF: 12.3 h; SGF: 13.9 h; each *p* < 0.001 vs. IGF).

### 2.2. Biomarker Kinetics during the First Week after Renal Transplantation

[TIMP-2] × [IGFBP7] levels were significantly different between DGF and SGF, as well as IGF, as early as day 1 after kidney transplantation (Figure 2) (median [TIMP-2] × [IGFBP7] in (ng/mL)^2^/1000: DGF 1.55 (range 0.02–15.20); SGF 0.49 (range 0.06–12.98); IGF 0.17 (range 0.02–32.17)). Median [TIMP-2] × [IGFBP7] levels were different between DGF and IGF, as well as SGF, up to day 5 after renal transplantation.

Serum creatinine was unable to differentiate between DGF and SGF at days 1–4 after renal transplantation (Figure 3) (median serum creatinine at day 1 in mg/dL: DGF 5.7 (range 3.0–9.9); SGF 5.75 (range 2.53–9.60); IGF 4.2 (range 0.81–9.75)). Renal function improvement in SGF was first detectable at day 5 compared to the DGF group (median serum creatinine at day 5 in mg/dL: DGF 5.6 (range 1.6–12.8); SGF 3.9 (range 1.4–9.3); IGF 1.7 (range 0.7–7.6)).

### 2.3. [TIMP-2] × [IGFBP7] before Renal Transplantation

Urinary [TIMP-2] × [IGFBP7] levels were measured in all patients able to provide urine before transplantation (79 of 186 patients; 42.5%). Urinary [TIMP-2] × [IGFBP7] levels were available in 49.5% of patients later showing IGF and 44.2% of patients later showing SGF. Only six patients (17.6%) of those later classified as having DGF were able to provide urine. Urinary [TIMP-2] × [IGFBP7] levels before KTx were significantly increased in patients later developing DGF in comparison to those showing IGF or SGF (Figure 4).

### 2.4. [TIMP-2] × [IGFBP7] Kinetics during the First 48 h after Renal Transplantation

In a subgroup of ten patients, urinary [TIMP-2] × [IGFBP7] levels were measured early after the transplant procedure every 6 h up to 48 h post-transplant in order to detect early changes. Two out of three patients later classified as DGF showed urinary [TIMP-2] × [IGFBP7]-levels > 1.2 (ng/mL)^2^/1000 as early as 6 h post-transplant, and all three patients were constantly above this threshold 24–48 h post-transplant. All other patients with either SGF (n = 2) or IGF (n = 5) revealed urinary [TIMP-2] × [IGFBP7]-levels <1.2 (ng/mL)^2^/1000 during the first 12–48 h post-transplant (Figure 5).

### 2.5. [TIMP-2] × [IGFBP7] Levels in Relation to CIT at Time of Renal Transplantation

A total of 53% of all patients had a CIT below 10 h, 42% had a value between 10 and 20 h, and only 5% had a value longer than 20 h. [TIMP-2] × [IGFBP7] levels were significantly higher in patients with a CIT between 10 and20 h compared to those with a CIT > 20 h on day 1 following KTx only. However, [TIMP-2] × [IGFBP7] levels did not differ based on CIT at all other time points (Figure 6A). In patients receiving a living donor transplantation, [TIMP-2] × [IGFBP7] levels were generally low but increased over time (Figure 6B). In patients receiving a living donor transplantation followed by IGF (43 of 48 patients), [TIMP-2] × [IGFBP7] was only significantly lower compared to patients with a CIT of 10–20 h on day 1 following KTx.

### 2.6. [TIMP-2] × [IGFBP7] Levels in Patients with Rejection

A total of 32 patients received a transplant biopsy for suspected rejection within the first 14 days post-transplant. A total of 20 rejection episodes were confirmed. Rejections were confirmed in 10 out of 15 biopsies (67%) performed in the DGF group, 3 out of 9 (33%) in the SGF group, and 7 out of 8 (88%) in the IGF group. [TIMP-2] × [IGFBP7] levels measured during the first 7 days post-transplant were not able to predict rejection.

### 2.7. [TIMP-2] × [IGFBP7] Levels in Relation to eGFR 3 Months after Transplantation

[TIMP-2] × [IGFBP7] levels were studied with regard to renal transplant function 3 months post-transplant. Three different functional groups were defined by eGFR at 3 months post-transplant: group A: eGFR 15 to 30 mL/min/m^2^; group B: eGFR 30 to 60 mL/min/m^2^; and group C: eGFR higher than 60 mL/min/m^2^. Interestingly median [TIMP-2] × [IGFBP7] levels were significantly higher at the first day post-transplant in patients with the lowest eGFR group compared to both of the other groups (Figure 7).

### 2.8. Performance of [TIMP-2] × [IGFBP7] for Identification of Non-DGF

ROC analysis of [TIMP-2] × [IGFBP7] at day 1 and day 2 post-transplant was comparable for the event “Non-DGF” (Figure 8). Three different cut-off values for [TIMP-2] × [IGFBP7] were analyzed, revealing a favorable test performance with a sensitivity of 87% and a specificity of 71% at a cut-off value of 0.9 (ng/mL)^2^/1000 at day 1 post-transplant. The positive predictive value for non-DGF was 93% for this cut-off value. A cut-off value of 2 (ng/mL)^2^/1000 resulted in a comparable high sensitivity of 93% with a lower specificity of 45%. A cut-off value of 0.3 (ng/mL)^2^/1000 showed a low sensitivity of 58%.

At day 2 post-transplant, test performance was most favorable with a sensitivity of 94% and a specificity of 53% at a cut-off value of 1.9 (ng/mL)^2^/1000. Indicators of performance at different cut-off values are shown in Table 2.

## 3. Discussion

Our prospective multicenter cohort study showed that post-transplant urinary [TIMP-2] × [IGFBP7] of the recipient can differentiate between renal transplant recipients with DGF and non-DGF as early as day 1 after transplantation. Renal function parameters were not able to differentiate between DGF and SGF early after KTx. ROC analysis of [TIMP-2] × [IGFBP7] at day 1 post-transplant for the event “Non-DGF” revealed a cut-off value of 0.9 (ng/mL)^2^/1000 with a positive predictive value of 93%.

The incidence of DGF increases due to the use of marginal kidneys necessitated by organ shortage, and is reported to be between 23 and 50% [11,12,13]. While the definition of SGF varies, it impacts allograft and clinical outcomes, leading to a reduced patient and graft survival [8,9,14]. The definition of SGF of a decrease of <10% in serum creatinine within the first 24 h following renal transplantation without need for dialysis allows the discrimination of patients with either slow or possibly even delayed graft function as early as day one post-KTx. In addition, using a percentage-based definition of SGF instead of absolute thresholds might relativize for individual levels of serum creatinine at time of transplantation.

Standard of care is the daily evaluation of diuresis, serum creatinine, and ultrasound of the renal transplant, plus varying clinical and laboratory parameters, by an experienced transplant physician. Identification of easily obtainable non-invasive biomarkers in addition to a standard of care that can predict changes in renal graft function is one of the major focuses of transplant medicine [15]. Ischemia and reperfusion injury occurring during transplantation triggers and upregulates several pathways [16]. Previous studies have demonstrated that specific biomarkers in both urine and blood can respond to glomerular or tubular injury. These may allow for additional information on graft function beyond routine clinical parameters.

The urinary biomarker combination [TIMP-2] × [IGFBP7], which are both markers for G1 cell cycle arrest, has been identified and validated to predict moderate to severe acute kidney injury in critically ill patients earlier than other known biomarkers [10]. Subsequent validations in several other clinical settings followed [17,18,19,20,21]. Given its commercial availability, the application of this test in the clinical setting is practicable. We therefore consciously omitted individual evaluation of TIMP-2 or IGFBP7, although TIMP-2 alone has been reported to show superior results in similar clinical settings [22,23]. The missing opportunity of individual evaluation of TIMP-2 or IGFBP7 does limit the interpretation of our data. While a correction of the urinary biomarker combination [TIMP-2] × [IGFBP7] for urine dilution is also not included in the commercially available test system, a correction for dilution to convert urine measurements from concentration to excreted amount/mass has been discussed [24,25]. If this is undertaken, the risk score has to be corrected by the squared dilution because two variables are multiplied that need to be individually corrected for dilution. Results might than be displayed as µg^2^/mmol^2^ [26]. Correction for dilution by dividing both values, TIMP-2 and IGFBP7, by the urinary creatinine concentration measured at the same time might add further information, especially early after KTx, where urine output, renal ability for concentration, and hydration status are variable. As a separate reporting of the two biomarkers is currently unavailable using the commercial test, normalization for urinary creatinine concentration was not performed.

TIMP-2 has a molecular size of approximately 21 kDa [27], while IGFBP7 has a molecular size of more than 23 kDa [28]. High-flux dialyzers used for intermittent hemodialysis in our centers are able to remove low- and middle-sized molecules with a molecular weight of 10–15 kDa [29]. Therefore, effects of intermittent hemodialysis on urinary [TIMP-2] × [IGFBP7] levels are not expected.

In the setting of transplantation, organs are generally exposed to conditions, especially ischemia and reperfusion, which are very likely to favor G1 cell cycle arrest and, thereby, disable cell division. It therefore seems very reasonable that cut-off values of urinary [TIMP-2] × [IGFBP7] for non-DGF are higher than those validated for the identification of acute kidney injury in critically ill adults [30]. Our results are in line with those of Schmitt et al. [31] and Pianta et al. [22], who also showed significantly increased urinary [TIMP-2] × [IGFBP7] levels in patients suffering from DGF following KTx.

Our kinetics indicated that measurement of [TIMP-2] × [IGFBP7] should be performed 24–48 h post-transplant. A single measurement within this time frame should be enough for identification of patients at risk of DGF. These results are in line with case reports [21] and the study of Schmitt et al., who also identified [TIMP-2] × [IGFBP7] at day 1 following transplantation as a marker for the development of DGF [31]. Measurement of [TIMP-2] × [IGFBP7] within 6 h post-transplantation was inferior to measurement at day 1 with regard to detection of non-DGF and a higher cut-off value was identified. This result is comparable with that of Yang et al., who identified [TIMP-2] × [IGFBP7] in immediately postoperative urine as a predictive marker of DGF in kidney transplantation, but with a higher cut-off of 1.39 (ng/mL)^2^/1000 and a reduced ROC of 86.7% [32].

[TIMP-2] × [IGFBP7] levels were also significantly increased in recipients later developing DGF in comparison to those classifying as SGF or IGF. Although reaching significance, values measured in different groups were overlapping and not predictive of individual outcomes. No correlation to specific diseases could be found. The meaningfulness of this result is also highly limited by the fact that only six patients of the DGF group (17.6%) were able to provide urine before transplantation, while 49.5% of the IGF group and 44.2% of the SGF group were available for measurement. Therefore, the existence of urinary output itself before KTx might reduce the risk of developing DGF, as anuric patients are more vulnerable to hyperkalemia or hyperhydration. While there are data showing that anuria is a risk factor for urological complications [33], studies on recovery of renal function following KTx in adults are missing.

[TIMP-2] × [IGFBP7] can differentiate kidneys with an increased CIT from those with a CIT value < 10 h. Cold storage decreases hypoxia-induced cell damage, but induces cellular injury by itself and injury is even further aggravated during reperfusion [16]. It therefore seems logical that increased CIT occurs as [TIMP-2] × [IGFBP7] values increase, as more cells go into cell cycle arrest. A CIT value > 20 h does not seem to further increase [TIMP-2] × [IGFBP7] values, but leads to prolonged proof of cell cycle arrest. Nevertheless, CIT alone was not able to predict DGF.

Elevated levels of [TIMP-2] × [IGFBP7] in patients receiving living kidney donation were associated with surgical difficulties during implantation. No clear association between elevated [TIMP-2] × [IGFBP7] in patients with deceased donors and CIT < 10 h and clinical data could be found, probably reflecting the limited knowledge on acute and long-term medical history of the donor.

[TIMP-2] × [IGFBP7] levels measured during the first 7 days post-transplant were not able to predict rejection within the first 2 weeks following transplantation. In a study by Elnokeety et al., acute rejection was associated with higher [TIMP-2] × [IGFBP7] levels than non-rejection-associated acute allograft dysfunction [34]. In contrast to our study, acute allograft dysfunction was not studied immediately post-transplant, but renal function had already been allowed to stabilize for 1–6 months with a baseline serum creatinine level of less than 1.5 mg/dL.

As in our study, where an eGFR 15–30 mL/min/m^2^ at 3 months post-transplant was associated with increased early [TIMP-2] × [IGFBP7] values, Kashani et al. and Hoste et al. showed a correlation of early [TIMP-2] × [IGFBP7] elevation with reduced clinical outcome during follow-up [10,30].

[TIMP-2] × [IGFBP7] measured at day one post-transplant was able to discriminate between DGF and non-DGF. Detecting patients with high probability of DGF early on may enable the implementation of innovative interventions. The use of [TIMP-2] ^×^ [IGFBP7] in the evaluation of reconditioning treatments such as machine reperfusion would also be conceivable [35]. However, comparison of findings with other studies is limited due to inhomogeneous definitions of DGF, SGF, and IGF. Measuring [TIMP-2] × [IGFBP7] in deceased donors before organ donation might offer additional information [36], but availability of samples is very limited.

In conclusion, early measurement of [TIMP-2] × [IGFBP7] can help to optimize patient selection for diagnostic or therapeutic actions in the early post-transplant phase and thereby protect patients’ personal and clinical resources.

## 4. Materials and Methods

### 4.1. Patient Population

We performed a prospective multicenter observational cohort study in all kidney transplant recipients older than 18 years and eligible for inclusion between November 2014 and July 2017 at the University Hospital Bern, Switzerland and the University Hospital Essen, Germany.

The study was performed in accordance with the Declaration of Helsinki and the Good Clinical Practice guidelines of the International Conference on Harmonization. The study protocol was approved by the local ethics committee of the University Duisburg-Essen (14-6153-BO) and the ethics committee of the canton Bern (Number: 098/14). All patients gave written informed consent to participate in the study before enrolment.

We included recipients receiving living or deceased donor transplantations, without any preselection based on pre-transplant, donor criteria, or the immunization status of the recipient. Only eligible transplant recipients with urine samples on at least 5 out of 7 days during the first week after renal transplantation were considered for the final analysis. Patients who underwent bladder lavage with solutions containing methylene blue were excluded due to potential interference with study parameters.

All patients received an initial immunosuppressive regimen as per the local standard of care (Table 1). All patients received prophylactic treatment with 960 mg of trimethoprim/sulfamethoxazole two to three times a week, at least during the first 6 months after transplantation. Furthermore, all patients at high risk for cytomegalovirus infection (donor serologically positive and recipient serologically negative) received valganciclovir in a prophylactic dosage for at least 6 months.

### 4.2. Definition of Graft Function during the First Week after Renal Transplantation

Early graft function after renal transplantation was graded in three different categories. Delayed graft function (DGF) was defined by the need for dialysis in the first week after renal transplantation. In the remaining group of patients, those transplant recipients with an immediate graft function (IGF) defined by a decrease of ≥10% in serum creatinine within the first 24 h following renal transplantation were separated from those with a slow graft function (SGF) defined by a decrease of <10% in serum creatinine within the first 24 h following renal transplantation without need for dialysis [37].

### 4.3. Urine Sample and Data Collection

Urine was collected at postoperative day 1 to day 7 daily following the same standard operating procedure throughout the study in both centers. In a subgroup of ten consecutive renal transplant recipients, urine samples were collected every 6 h during the first 24 h and at 36 and 48 h to monitor early kinetics of post-transplant urinary [TIMP-2] × [IGFBP7]. Routine laboratory parameters were monitored and documented daily from postoperative day 1 to day 7, at 2 weeks, 3 months, and 12 months postoperatively as part of the clinical routine management. They were measured by the central laboratory units of the University Hospital Essen or the University Hospital Bern. In addition, clinical events such as need for dialysis during the first postoperative week or rejection episodes during the first two weeks after renal transplantation were documented. 

### 4.4. Measurement of TIMP-2 and IGFBP7

The biomarker combination [TIMP-2] × [IGFBP7] in urine was analyzed using the commercially available and FDA-approved NephroCheck^®^ (Astute Medical, San Diego, CA, USA/bioMérieux, France). The ASTUTE140^®^ meter, a bench-top analyzer, detects fluorescent signals from the immunoassays of TIMP-2 and IGFBP7 and converts the fluorescent signal obtained from each of the two biomarker assays into a single numerical result, the AKIRisk^®^ Score. The AKIRisk^®^ Score expresses ([TIMP-2] × [IGFBP7])/1000. Results are displayed as (ng/mL)^2^/1000, range from 0.04 to 10.0, and are obtained within 20 min.

Quality controls and calibration were performed as recommended and provided by the manufacturer.

Urine samples were either measured within 4 h after collection or frozen at −20 °C and stored until measurement. Frozen samples were thawed at room temperature and the NephroCheck^®^ was performed immediately.

### 4.5. Statistical Analysis

Statistical comparisons were performed using the chi-square test and Fisher’s exact test for categorial data or Kruskal–Wallis test for non-parametric data followed by Dunn’s multiple comparisons test. For all analyses, two-sided *p*-values of less than 0.05 were considered statistically significant. Receiver operating characteristic (ROC) curves were generated to analyze biomarker performances. The optimal cut-off level was defined by the highest Youden index. Statistical analysis was performed using SPSS Statistics 26 (IBM, Armonk, NY, USA) and Graph Pad Prism 8 (Graph Pad Software, La Jolla, CA, USA).

## Figures and Tables

**Figure 1 ijms-25-04169-f001:**
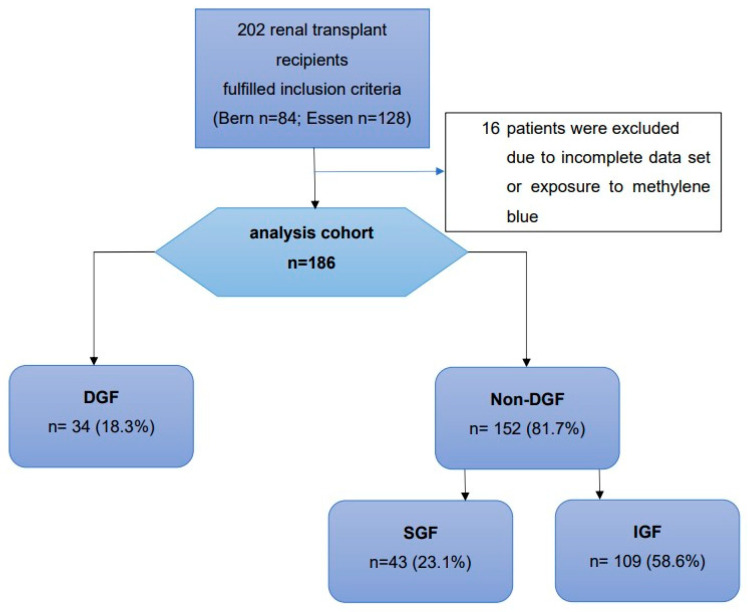
Study flow chart. *DGF*, delayed graft function; *SGF*, slow graft function; *IGF*, immediate graft function.

**Figure 2 ijms-25-04169-f002:**
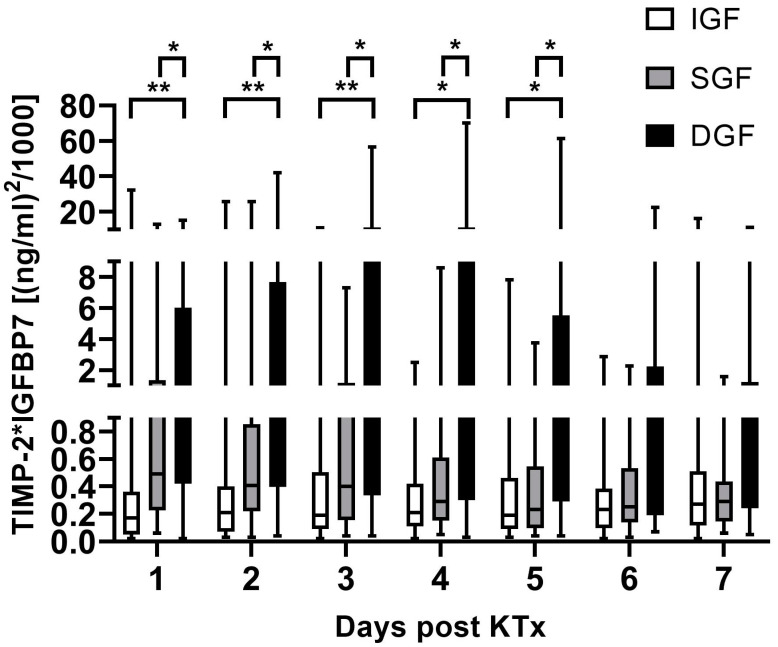
[TIMP-2] × [IGFBP7] levels at days 1 to 7 following kidney transplantation grouped by allograft function recovery. * *p* < 0.05 and ** *p* < 0.01. *DGF*, delayed graft function; *SGF*, slow graft function; *IGF*, immediate graft function; *KTx*, kidney transplant.

**Figure 3 ijms-25-04169-f003:**
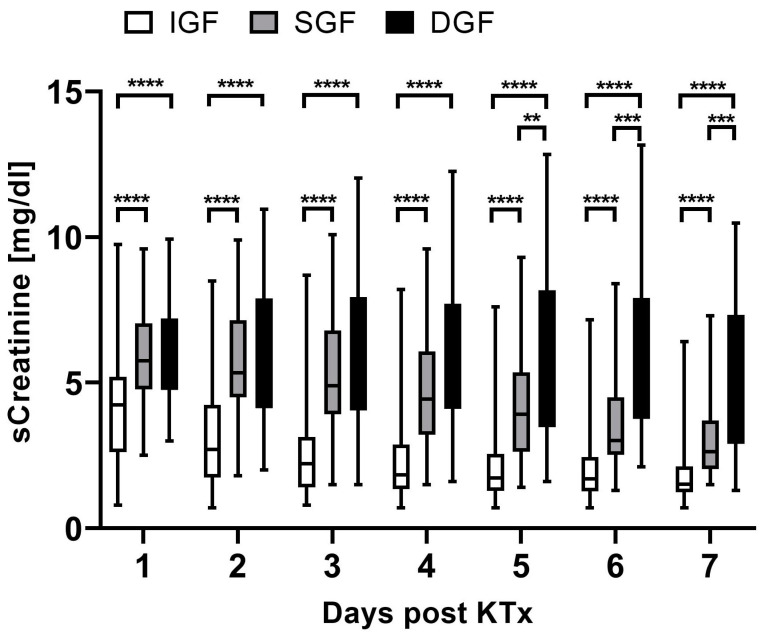
Serum creatinine levels at days 1 to 7 following kidney transplantation grouped by allograft function recovery. ** *p* < 0.01, *** *p* < 0.001 and **** *p* < 0.0001. *DGF*, delayed graft function; *SGF*, slow graft function; *IGF*, immediate graft function; *KTx*, kidney transplant; *sCreatinine*, serum creatinine.

**Figure 4 ijms-25-04169-f004:**
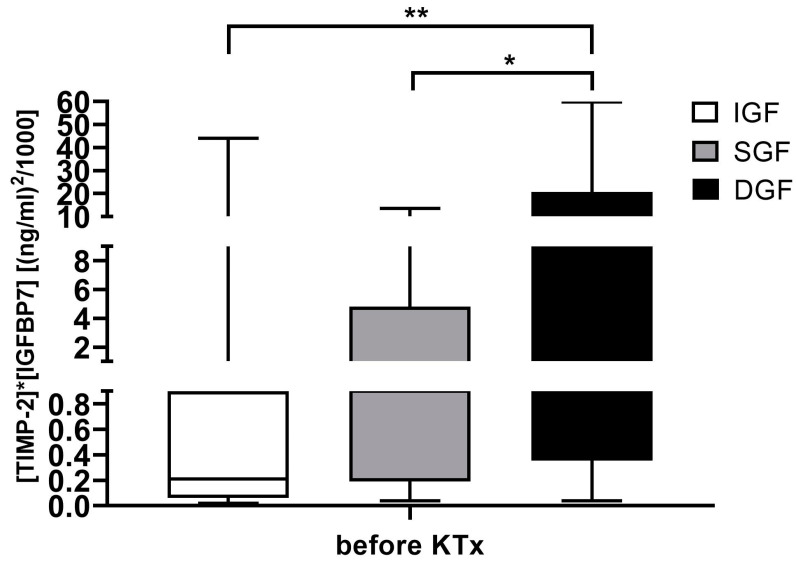
[TIMP-2] × [IGFBP7] levels before kidney transplantation grouped by allograft function recovery. * *p* < 0.5 and ** *p* < 0.01. *DGF*, delayed graft function; *SGF*, slow graft function; *IGF*, immediate graft function; *KTx*, kidney transplant.

**Figure 5 ijms-25-04169-f005:**
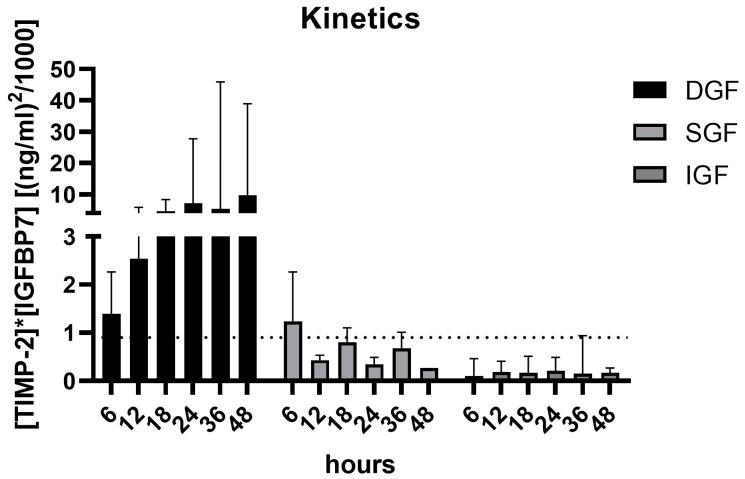
[TIMP-2] × [IGFBP7] kinetics during the first 48 h in a subgroup of 10 patients after renal transplantation grouped by allograft function recovery. *DGF*, delayed graft function (n = 3); *SGF*, slow graft function (n = 2); *IGF*, immediate graft function (n = 4). Dashed line marks a cut-off value of 0.9 (ng/mL)^2^/1000 for prediction of Non-DGF.

**Figure 6 ijms-25-04169-f006:**
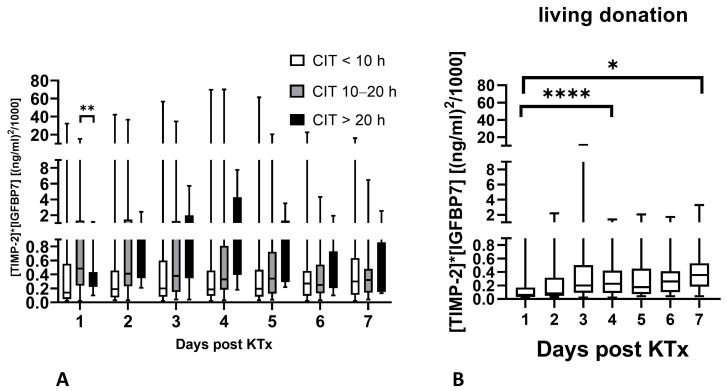
[TIMP-2] × [IGFBP7] levels in relation to cold ischemic time (**A**) and [TIMP-2] × [IGFBP7] levels after living kidney donation (**B**). * *p* < 0.05, ** *p* < 0.01, and **** *p* < 0.0001. *CIT*, cold ischemic time; *KTx*, kidney transplant.

**Figure 7 ijms-25-04169-f007:**
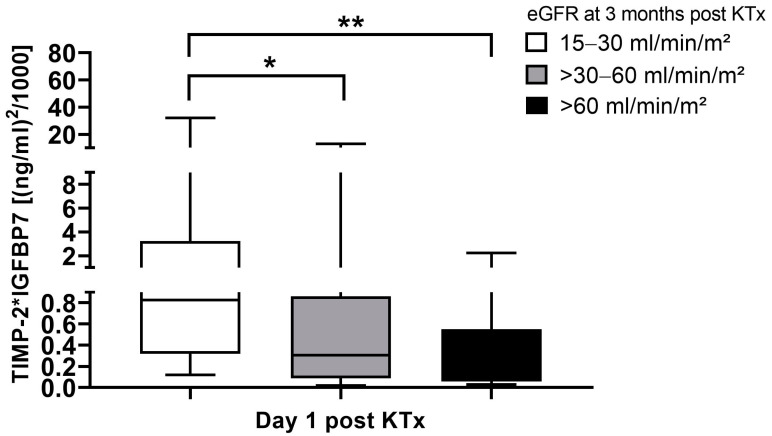
[TIMP-2] × [IGFBP7] levels at day 1 post-KTx in relation eGFR 3 months post-KTx. * *p* < 0.05 and ** *p* < 0.01. *eGFR*, estimated glomerular filtration rate; *KTx*, kidney transplant.

**Figure 8 ijms-25-04169-f008:**
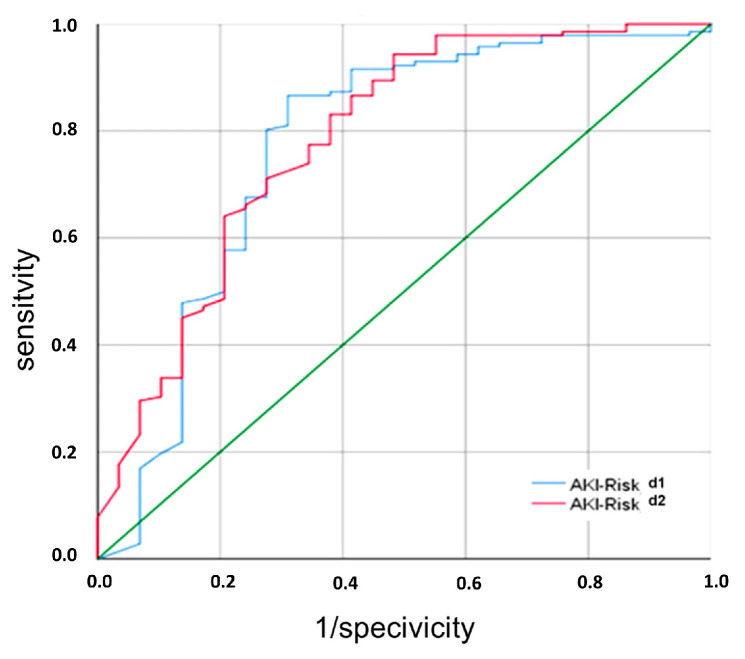
ROC curve for prediction of non-delayed graft function by urinary [TIMP-2] × [IGFBP7] levels at days 1 and 2 following renal transplantation. *AKI*, acute kidney injury; *d*, day.

**Table 1 ijms-25-04169-t001:** Patient characteristics. *DGF*, delayed graft function; *SGF*, slow graft function; *IGF*, immediate graft function; *CIT*, cold ischemic time; *ADPKD*, autosomal dominant polycystic kidney disease; *aHUS*, atypical hemolytic uremic syndrome; *ns*, not significant; *BMI*, body mass index; *PRA*, panel reactive antibodies; *IL-2*, interleukin 2; *ATG*, anti-thymocyte globulin.

	Patient Population	DGF	SGF	IGF	
	(n = 186)	(n = 34)	(n = 43)	(n = 109)	*p*-Value
**recipient characteristics**					
female, n (%)	73 (39)	14 (41)	14 (33)	45 (41)	ns
renal re-transplants, n (%)	32 (17)	6 (18)	5 (12)	21 (19)	ns
Age [years], median (range)	55 (18–78)	56 (25–70)	58 (30–74)	51 (18–78)	0.011
BMI [kg/m^2^], median (range)	25 (17–43)	28 (19–38)	26 (21–43)	24 (17–41)	<0.001
Dialysis vintage [in months], median (range)	32 (0–184)	45 (2–139)	38 (0–125)	20 (0–184)	<0.001
PRA [%], median (range)	8 (0–100)	13 (0–100)	11 (0–100)	6 (0–100)	ns
PRA 0–5%, n (%)	156 (84)	27 (79)	34 (79)	95 (87)	ns
PRA 5–20%, n (%)	9 (5)	0	4 (9)	5 (5)	ns
PRA > 20%, n (%)	21 (11)	7 (21)	5 (12)	9 (8)	ns
**cause for CKD**					
diabetic nephropathy, n (%)	16 (9)	5 (15)	5 (12)	6 (6)	ns
vascular nephropathy, n (%)	12 (6)	1 (4)	3 (7)	8 (7)	ns
ADPKD, n (%)	39 (21)	9 (26)	8 (19)	22 (20)	ns
glomerulonephritis, n (%)	68 (36)	10 (29)	18 (42)	40 (37)	ns
aHUS, n (%)	5 (3)	0	2 (4)	3 (3)	ns
others, n (%)	46 (25)	9 (26)	7 (16)	30 (28)	ns
**Initial immunosuppressive therapy**					
IL-2 Receptor antagonist, n (%)	181 (97)	34 (100)	41 (95)	106 (97)	ns
ATG, n (%)	5 (3)	0	2 (5)	3 (3)	ns
Cyclosporin A, n (%)	64 (34)	13 (38)	6 (14)	45 (41)	0.005
Tacrolimus, n (%)	122 (66)	21 (62)	37 (86)	64 (59)	0.005
Azathioprin, n (%)	3 (2)	0	1 (2)	2 (2)	ns
Mycophenol mofetil/-acid, n (%)	175 (94)	33 (97)	41 (95)	101 (93)	ns
Everolimus, n (%)	8 (4)	1 (3)	1 (2)	6 (6)	ns
Steroids, n (%)	186 (100)				ns
**donor characteristics**					
living donation, n (%)	48 (26)	3 (9)	2 (5)	43 (39)	<0.001
female, n (%)	102 (55)	19 (56)	20 (47)	63 (58)	ns
Age [years], median (range)	54 (0–86)	54 (22–78)	61 (8–86)	53 (0–85)	0.018
CIT [hours], median (range)	8.9 (0.8–29.2)	12.3 (2.6–29.2)	13.9 (2.7–21.8)	6.3 (0.8–24.3)	<0.001
CIT < 10 h, n (%)	99 (53)	12 (35)	13 (30)	74 (68)	<0.001
CIT >10 h, n (%)	87 (47)	22(65)	30 (70)	35 (32)	<0.001

**Table 2 ijms-25-04169-t002:** Performance of [TIMP-2] × [IGFBP7] at day 1 and day 2 post-transplant at different cut-off values.

[TIMP-2] × [IGFBP7] Cut-Off [(ng/mL)^2^/1000]	Sensitivity	Specificity	Positive Predictive Value	Negative Predictive Value
Post-transplant Day 1				
0.3	58%	81%	93%	29%
**0.9**	**87%**	**71%**	**93%**	**54%**
2.0	93%	45%	89%	58%
Post-transplant Day 2				
0.3	58%	81%	93%	29%
0.9	85%	63%	91%	47%
**1.9**	**94%**	**53%**	**90%**	**65%**
2.0	94%	50%	90%	64%

Favourable cut-off values are high-lighted in bold.

## Data Availability

The datasets used and/or analyzed during the current study are available from the corresponding author on reasonable request.

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
