# Peer review of "Urinary Biomarkers for Cell Cycle Arrest TIMP-2 and IGFBP7 for Prediction of Graft Function Recovery after Kidney Transplantation"

_ijms, 2024, doi:10.3390/ijms25084169_

Round 1
Reviewer 1 Report
Comments and Suggestions for Authors
Thank you for the opportunity to review this work by Gackler and colleagues! The topic is of interest. The manuscript is well written. I have some concerns: We are only presented with data on the combination of u-TIMP-2 and IGFBP7 for prediction of DGF. It would be interesting to see data on individual markers and discuss any discrepancies in light of findings in Pianta et al (ref 22) and Schmitt et al (ref 25). Do they consider their findings as confirmatory of findings in ref 22 and 25? Why did they restrict markers to TIMP-2 and IGFBP7 and not include e.g. VEGF as shown in ref 22? Why did they use (ng/ml)2 / 1000 instead of ng/mmol creatinine?
Reviewer 2 Report
Comments and Suggestions for Authors
If DGF can be predicted by measuring urinary TIMP-2*IGFBP7 during early phase after kidney transplantation, it could be very useful because it will lead to decision making in which hemodialysis can be performed quickly without the need for conservative management such as pressor treatment, blood transfusion, and addition of fluid replacement.
This is a valuable paper for readers, however, there are several points that should be considered more extensively.
Is it OK to ignore urinary TIMP-2*IGFBP7 levels before kidney transplantation in the recipients with urine output?
I believe there were cases of high levels of TIMP-2*IGFBP7 even in kidney transplants from living donors or deceased donors with short CIT, but were there any characteristics?
Is there any relationship between TIMP-2*IGFBP7 levels and the pathological findings of the 0hr biopsy.
Reviewer 3 Report
Comments and Suggestions for Authors
In this pilot study, the authors show the diagnostic potential of TIMP-2 and IGFBP7 for graft function recovery in patients after kidney transplantation. The structure of the paper is easy to read, and the content is appropriate. There are only a few minor questions;
1. SGF is intermediate between IGF and DGF, and definitions vary, such as Cr ≥ 1.5-3.0 mg/dL on POD5-14 without dialysis. What is the rationale for “a decrease of <10 % in serum creatinine within the first 24 hours following renal transplantation without need for dialysis”?
2. Please graphically show the relationship between eGFR and biomarkers three months after transplantation, if possible.
